# The P2X4 Receptor: Cellular and Molecular Characteristics of a Promising Neuroinflammatory Target

**DOI:** 10.3390/ijms23105739

**Published:** 2022-05-20

**Authors:** Reece Andrew Sophocleous, Lezanne Ooi, Ronald Sluyter

**Affiliations:** 1Illawarra Health and Medical Research Institute, Wollongong, NSW 2522, Australia; reeces@uow.edu.au (R.A.S.); lezanne@uow.edu.au (L.O.); 2Molecular Horizons and School of Chemistry and Molecular Bioscience, University of Wollongong, Wollongong, NSW 2522, Australia

**Keywords:** P2X4 receptor, *P2RX4* gene, purinergic receptor, purinergic signalling, extracellular ATP, inflammation, neuroinflammation, pain, macrophage, microglia

## Abstract

The adenosine 5′-triphosphate-gated P2X4 receptor channel is a promising target in neuroinflammatory disorders, but the ability to effectively target these receptors in models of neuroinflammation has presented a constant challenge. As such, the exact role of P2X4 receptors and their cell signalling mechanisms in human physiology and pathophysiology still requires further elucidation. To this end, research into the molecular mechanisms of P2X4 receptor activation, modulation, and inhibition has continued to gain momentum in an attempt to further describe the role of P2X4 receptors in neuroinflammation and other disease settings. Here we provide an overview of the current understanding of the P2X4 receptor, including its expression and function in cells involved in neuroinflammatory signalling. We discuss the pharmacology of P2X4 receptors and provide an overview of P2X4-targeting molecules, including agonists, positive allosteric modulators, and antagonists. Finally, we discuss the use of P2X4 receptor modulators and antagonists in models of neuroinflammatory cell signalling and disease.

## 1. Introduction

The P2X4 receptor, encoded by the *P2RX4* gene, is a trimeric adenosine 5′-triphosphate (ATP)-gated cation channel that is expressed primarily in cells and tissues of the immune and central nervous systems where it has established roles primarily in the regulation of neuroinflammation [1,2]. In the cells of the immune system, the activation of P2X4 receptors by ATP results in the influx of cations and downstream inflammatory signalling events, such as the release of prostaglandin E_2_ from macrophages [3], release of brain-derived neurotrophic factor (BDNF) from microglia [4], and blood flow-dependent Ca^2+^ signalling in endothelial cells [5,6]. As such, changes in P2X4 receptor expression or function have been associated with the transmission of inflammatory and neuropathic chronic pain [7,8] and cardiovascular disease [9], as well as alcohol use disorders [10], neuropsychiatric disorders [11], and alterations in neuroplasticity [12].

P2X4 receptors were first cloned from rat [13] and human [14] brain complementary DNA, revealing a protein 388 amino acids in length. Studies of the human P2X1 and rat P2X2 receptors have suggested that the P2X4 receptor could contain up to ten disulphide-bonding cysteine residues in the extracellular loop region [15,16]. These cysteine-cysteine interaction properties were later confirmed when the first high resolution (<3 Å) crystal structure of the (zebrafish) P2X4 receptor was reported in the ATP-unbound, closed state [17] and soon after in the ATP-bound, open state (Figure 1A) [18]. These studies revealed the ‘dolphin-like’ shape of P2X4 monomeric subunits, with the ‘tail fluke’ submerged in the lipid membrane and extending out into the cytoplasmic region of the cell and the ‘body’, including ‘dorsal fin, flippers, and head regions’, extruding out into the extracellular (or hydrophobic) region (Figure 1B). Although X-ray crystallography of the zebrafish P2X4 receptor utilised truncated forms, which lacked the N- and C- terminus, these truncated receptors still demonstrated functional P2X4 channel activity [17]. With these studies confirming the trimeric structure of P2X4 receptors, researchers have been able to identify key roles for disulphide-bonding cysteine residues in P2X4 receptor ligand-binding pocket formation and channel gating [19]. Despite a strong push towards understanding the structural biology of the P2X4 receptor, the expected development of effective and selective molecular tools to study the role of P2X4 receptors in physiology and pathophysiology has not been as successful as for other purinergic receptors of similar structure, such as the P2X3 or P2X7 receptors. In this review, we will give an overview of the cell and molecular characteristics of P2X4 receptors, including their expression and function. We detail the agonists, antagonists, and modulators of P2X4 receptors and provide an update on developments of novel P2X4 receptor-targeting molecules. Finally, we discuss advances in understanding the role of P2X4 receptors in neuroinflammatory physiology and pathophysiology.

## 2. P2X4 Receptor Expression and Function

The P2X4 receptor was originally detected in rat [13,21] and human brain tissue [14]. Functional P2X4 receptors have since been reported in a wide distribution of mammalian cells and tissues [22] (Figure 2), including those of the central nervous system [23,24,25,26,27] and in peripheral endothelial cells, where they play an important role in the regulation of vascular tone [5,6,28]. P2X4 receptors have also been identified in a wide range of mammalian immune cells [29], including B lymphocytes [30], as well as T lymphocytes, where P2X4 receptor activation has been attributed to ATP release and autocrine signalling, mitochondrial metabolism, cell polarisation, and cell migration [31,32,33]. P2X4 receptors are also expressed on mast cells [34], where they are involved in mast cell degranulation [35], as well as on other myeloid cells including monocytes [36,37,38], tissue-resident macrophages [3,39], and eosinophils [40]. P2X4 receptors are also commonly observed on cultured mammalian monocyte and macrophage cells and differentiated cell lines [39,41,42,43,44,45,46]. P2X4 receptors expressed on microglia have also been well-documented, and together with macrophages, have known roles in signalling pathways that mediate neuroinflammatory responses and chronic pain [3,4,24,47,48,49]. Finally, it should be noted that P2X4 receptors are also widely expressed in central and peripheral neurons and play important roles in neurotransmission (as reviewed in [2,50]). Notably, the P2X4 knockout mouse showed alterations in synaptic potentiation [51]. Neuronal P2X4 receptors are thus likely to be involved in neuroinflammation and chronic pain directly and potentially indirectly via activation of microglia [52].

P2X4 receptors in human and rodent macrophages, microglia, and endothelial cells are reported to be functional at the cell surface and intracellularly [53,54] (Figure 3), where they are targeted to lysosomal compartments through N-terminal di-leucine (L_22_I_23_) and C-terminal tyrosine-based (Y_372_XXV and Y_378_XXGL) motifs [54,55,56]. These motifs control the constitutive, dynamin-dependent internalisation and recycling of P2X4 receptors [57,58], while disruption of these targeting motifs resulted in an increase in the surface expression of P2X4 receptors [56]. The P2X4 receptor contains seven *N*-linked glycosylation sites which further aid in establishing expression at the cell surface [59] and in trafficking to and maintaining function within lysosomal compartments, while also resisting degradation [54,60]. Luminal pH within lysosomes has been demonstrated to have regulatory effects on P2X4 receptors [55]. It is in the lysosomes, as well as in late-endosomal compartments, where P2X4 receptors are reported to control a number of important physiological roles, such as Ca^2+^ regulation, lysosomal fusion, and receptor re-sensitisation [61]. This may also suggest a role for P2X4 receptors in autocrine signalling events such as lysosomal exocytosis, ATP release, and Ca^2+^ homeostasis, as has been established for other P2 receptors, such as the P2Y_2_ receptor [62,63].

## 3. Agonists of the P2X4 Receptor

The primary agonist of the human P2X4 receptor, as with all other mammalian P2X receptor subtypes, is ATP [64,65], with varying EC_50_ values reported between 0.2 and 10 μM (Table 1). Based on the data presented in Table 1, these differences most likely reflect varying assay conditions, rather than differences between P2X4 receptors from different species, with the exception of the zebrafish P2X4 receptor. Consistent with other P2X receptor subtypes, the activation of P2X4 receptors results in the influx of cations, such as Ca^2+^ and Na^+^, although P2X4 receptors are most permeable to Ca^2+^ [66]. Following activation, P2X4 receptors undergo relatively moderate desensitisation, which is dependent on two key residues: Lys373 and Tyr374 [67]. Pharmacological studies of cloned P2X4 receptors from rat brains reported maximal activation of the receptor via ATP-induced currents, with similar EC_50_ values [13,21,23]. These rat P2X4 receptors were also activated by other nucleotide analogues in the following order of decreasing potency: ATP > adenosine 5′-*O*-(3-thiotriphosphate) (ATPγS) > 2-methylthio-ATP (2MeSATP) >> adenosine 5′-diphosphate (ADP) ≈ α,β-methylene-ATP (α,β-meATP) [13,23]. 

The agonist profile for the human P2X4 receptor expressed in *Xenopus laevis* oocytes was found to be similar to that of the rat P2X4 receptor (ATP > 2MeSATP ≥ cytidine-5′-triphosphate (CTP) > α,β-meATP) [14]. Recombinant P2X4 receptor orthologues expressed in mammalian cell lines, including human, rat, mouse, dog, bovine, zebrafish, and *Xenopus* P2X4 receptors have since demonstrated sensitivity to ATP consistent with the low micromolar range determined in the original studies [68,69,70,71]. Studies have also demonstrated partial agonist activity of 3′-*O*-(4-benzoyl)benzoyl adenosine-5′-triphosphate (BzATP) and diadenosine polyphosphates (AP_4_A and AP_5_A), but not ADP at mammalian P2X4 receptors [69,70,71,72] The response of mammalian P2X4 receptors to ADP in some, but not all, studies has been attributed to the presence of traces of contaminating ATP in commercial ADP stocks [69].

**Table 1 ijms-23-05739-t001:** Activity of ATP in cells expressing different P2X4 receptor orthologues.

Target ^1^	EC_50_
hP2X4	0.2 µM [69], 0.5 µM [71], 0.7 µM [72], 1.4 µM [70], 5.0 µM [73], 7.4 µM [14], 10.2 µM [74]
rP2X4	1.7 µM [72], 2.3 µM [75], 4.1 µM [74,76], 5.5 µM [70], 6.9 µM [21], 7.9 µM [73], 10.0 µM [13,23]
mP2X4	0.3 µM [72], 1.7 µM [74], 2.3 µM [70], 6.3 µM [73]
dP2X4	0.3 µM [69]
cP2X4	9.5 µM [77]
zP2X4	274 µM [78]

^1^ hP2X4, human P2X4; rP2X4, rat P2X4; mP2X4, mouse P2X4; dP2X4, dog P2X4; cP2X4, chicken P2X4; zP2X4, zebrafish P2X4.

## 4. Positive Modulators of the P2X4 Receptor

A number of positive modulators of P2X4 receptors have been described with a range of therapeutic benefits [79]. Ivermectin is a broad-spectrum anti-parasitic drug that is commonly used in human and veterinary medicine [80]. The application of extracellular ivermectin (<10 µM) potentiated human and rodent P2X4 receptor-mediated currents and delayed channel deactivation [39,81,82]. ATP-induced currents potentiated by the application of extracellular ivermectin led to the identification of a binding pocket within the transmembrane domains of P2X4 receptors, which was found to play a key role in ivermectin recognition, binding, and selectivity [83,84,85]. Ivermectin has also demonstrated the potentiation of ATP-induced Ca^2+^ responses through native P2X4 receptors on human monocyte-derived macrophages [41,43] and human THP-1 monocytes [46]. Interestingly, given the use of ivermectin in veterinary medicine, this compound can also potentiate canine P2X4 receptors [69], but whether ivermectin has off-target effects in dogs remains unknown. However, breeds such as collies and sheepdogs, due to mutations in the *MDR1* gene and associated defects in the blood-brain barrier, are prone to ivermectin-induced neurotoxicity [86]. Other members of this family of lipophilic compounds, known as avermectins, including abamectin and moxidectin, have also demonstrated the potentiation of P2X4 receptors [87,88]. Although the sensitivity of ivermectin is greatest for P2X4 receptors, ivermectin can also potentiate human, but not rodent, P2X7 receptors [89]. 

Recently, a number of novel positive allosteric modulators of P2X4 receptors were identified from protopanaxadiol ginsenoside extracts of the Chinese medicinal plant, *Panax ginseng* [90]. These compounds, known as Rd and compound K (CK), demonstrated a two-fold potentiation of ATP-induced responses in cells expressing human P2X4 receptors across a range of techniques, including electrophysiology, Ca^2+^ flux assays, and fluorescent dye uptake assays [90]. Despite this, these ginsenosides can also potentiate ATP-induced responses at human and rodent P2X7 receptors [90,91,92,93].

Cibacron blue, an isomer of reactive blue 2 and a broad-spectrum P2X receptor inhibitor, demonstrated positive modulation of the rat P2X4 receptor at concentrations between 3 and 30 µM [94]. However, cibacron blue did not potentiate ATP-induced responses in cells expressing the human P2X4 receptor [14].

## 5. Antagonists of the P2X4 Receptor

### 5.1. Broad-Spectrum and Non-Selective Antagonists

Human P2X4 receptors, but not rat P2X4 receptors, display sensitivity to the broad-spectrum P2 receptor antagonists, pyridoxalphosphate-6-azophenyl-2′,4′-disulfonic acid (PPADS), and suramin when expressed in *Xenopus* oocytes [14]. In contrast, others have found PPADS to be a poor inhibitor of ATP-induced Ca^2+^ responses through recombinant human P2X4 receptors expressed in 1321N1 cells [71]. Likewise, PPADS and another broad-spectrum P2 receptor antagonist, reactive blue 2, (both up to 50 µM) failed to inhibit ATP-induced currents in *Xenopus* oocytes expressing the rat P2X4 receptor [13], with others even observing potentiation by these compounds in oocytes expressing the mouse P2X4 receptor [74]. This lack of inhibition by PPADS was later found to be restored upon replacing glutamic acid at position 249 with lysine, which is found at the equivalent position in PPADS-sensitive P2X receptors [23]. A similar effect was also observed with a single point mutation on the rat P2X4 receptor (Gln78Lys) resulting in an increased sensitivity to suramin [14]. Other non-selective P2 receptor antagonists, such as bromophenol blue and cibacron blue, were found to inhibit ATP-induced currents at the human and rat P2X4 receptor, although with greatly different species selectivity [14]. Similar findings have been observed with Brilliant Blue G, which can impair human, but not rat, P2X4 receptors at micro-molar concentrations, despite inhibiting both human and rat P2X7 receptors [95]. Ethanol has also demonstrated the inhibition of ATP-induced currents through rodent P2X4 receptors, which in itself can be antagonised by ivermectin [96,97].

High concentrations of extracellular Zn^2+^ and Cd^2+^ (>100 µM) have been observed to inhibit the rat P2X4 receptor [14,98], while at lower concentrations (10 µM) these divalent cations can potentiate ATP-induced currents. In contrast, Cu^2+^ and Hg^2+^ have demonstrated the time- and concentration-dependent inhibition of ATP-induced currents through the rat P2X4 receptor [98,99]. More recently, molecular dynamic simulations have revealed that Mg^2+^ complexed with ATP maintains the open channel state of the P2X4 receptors and that this process can be reversed when Mg^2+^ is exchanged with K^+^ [100]. In addition, rat and human P2X4 receptor activity are also modulated by extracellular H^+^, where decreases in pH (<6.5) inhibit P2X4 receptor activity, while increases above physiological pH potentiate channel activity [76,101]. The biological significance of pH-mediated regulation of the P2X4 receptor is supported by its lysosomal distribution [54], where the resting luminal pH of lysosomes (pH 4.6) can tightly regulate P2X4 receptor activity [102].

The broad-spectrum P2X receptor antagonist 2′,3′-*O*-(2,4,6-trinitrophenyl)adenosine-5′- triphosphate (TNP-ATP) [103] has demonstrated competitive inhibition of ATP-induced Ca^2+^ responses and inward currents through recombinant mammalian P2X4 receptors expressed in HEK293 and 1321N1 cells [69,72,104,105]. Nonetheless, TNP-ATP is a much more potent antagonist of P2X1, P2X2, and P2X3 heterotrimeric and homotrimeric receptors [106]. The selective serotonin reuptake inhibitor (SSRI) paroxetine has also been found to act as an allosteric antagonist of mammalian P2X4 receptors [69,107]. Similarly, the serotonin-norepinephrine reuptake inhibitor (SNRI) duloxetine also inhibited mammalian P2X4 receptors with similar selectivity and potency to paroxetine [69,108].

### 5.2. Selective Antagonists

Several more potent and selective P2X4 receptor antagonists have been identified and tested against the human and rodent P2X4 receptors. A chemical library screen identified 5-[3-(5-thioxo-*4H*-[1,2,4]oxadiazol-3-yl)phenyl]-*1H*-naphtho [1, 2-b][1,4]diazepine-2,4(*3H*,*5H*)-dione (NP-1815-PX) as a novel antagonist of mammalian P2X4 receptors [109]. A number of compounds derived from N-substituted phenoxazines have also been identified as selective P2X4 receptor antagonists [105], including the allosteric inhibitor *N*-(benzyloxycarbonyl)phenoxazine (PSB-12054), which blocked recombinant mammalian P2X4 receptor-mediated Ca^2+^ responses in 1321N1 cells [105]. Another study utilising the same expression model confirmed PSB-12054 as a potent antagonist of human the P2X4 receptor with a much greater selectivity for P2X4 compared to other P2X receptors [110]. Whilst PSB-12054 is amongst one of the more potent P2X4 receptor antagonists, it is poorly water-soluble, making it a difficult compound for both in vitro and in vivo use. An analogue of this inhibitor with greater water solubility, *N*-(p-methylphenyl)sulfonylphenoxazine (PSB-12062), was demonstrated to have a similar potency and selectivity to PSB-12054 at mammalian P2X4 receptors [105]. PSB-12062 has also demonstrated the inhibition of ATP-induced calcium responses in human THP-1 monocytes and differentiated macrophages [46] and the reduction of ATP-induced Ca^2+^ responses and CXCL5 secretion in human monocyte-derived macrophages [41,43]. 

The selective P2X4 receptor antagonist, 5-(3-bromophenyl)-1,3-dihydro-2*H*-benzofuro [3,2-*e*]-1,4-diazepin-2-one (5-BDBD) [111], was first reported to competitively inhibit ATP-induced currents and Ca^2+^ responses in HEK293 cells expressing the recombinant human P2X4 receptor [104]. Studies have also reported 5-BDBD as an antagonist of recombinant P2X4 receptors of other mammals, including dog, mouse, and rat P2X4 receptors [69,72,112]. Despite several studies indicating a competitive mechanism of antagonism, the mechanism of action of 5-BDBD is still of some debate. Using radioligand binding assays, it was demonstrated that 5-BDBD likely modulates P2X4 receptor activity through an allosteric site, rather than through interaction with the orthosteric binding site occupied by ATP [72]. This is further supported by molecular modelling and site-directed mutagenesis data, which indicate an allosteric binding region formed by the Met109, Phe178, Tyr300, and Ile312 residues of one P2X4 subunit, and the Arg301 residue of the adjacent subunit [113]. The aforementioned study also indicates a poor inhibitory effect on open or desensitising P2X4 receptors, suggesting that 5-BDBD likely acts as a negative allosteric modulator rather than a competitive antagonist as previously thought. Similar to PSB-12054, 5-BDBD has a low water solubility, which can severely hinder its suitability as both an in vitro and in vivo P2X4 receptor antagonist [114].

The phenylurea 1-(2,6-dibromo-4-isopropyl-phenyl)-3-(3-pyridyl)urea (BX430) was also identified as an allosteric inhibitor of the human and bovine, but not mouse or rat, P2X4 receptors and displayed strong selectivity compared with other P2X receptors [115]. BX430 also appears to partially inhibit canine, zebrafish, and *Xenopus* P2X4 receptor orthologues [68,69]. The selectivity of BX430 across a range of P2X4 receptor orthologues was demonstrated to be regulated by a single amino acid (Ile312) which forms a docking site with Asp88 and Tyr300, leading to reduced inhibitory effects at rodent and other non-mammalian P2X4 receptors [68].

Notably, the inhibition of endogenous P2X4 receptors on human or canine macrophages and macrophage cell models using selective antagonists such as PSB-12062, BX430, and 5-BDBD appears to have a limited effect on the peak calcium response but has a stronger inhibitory effect on the decay response kinetics and net calcium movement [41,42,43,46]. The incomplete inhibition of Ca^2+^ responses following treatment of these P2X4-expressing cells with thapsigargin or antagonists for other purinergic receptors indicates that P2X4 receptors are likely involved in co-signalling events with other ATP-activated P2 receptors.

Newly developed, yet relatively untested compounds have also emerged recently in an attempt to improve both the potency and the selectivity of P2X4 receptor antagonists. A potent, orally active, and selective P2X4 antagonist, *N*-[4-(3-chlorophenoxy)-3-sulfamoylphenyl]-2-phenylacetamide (BAY-1797), was discovered to have improved potency at human and rodent P2X4 receptors in vitro with little to no inhibitory effect on other P2X receptors [116]. This compound also demonstrated promising anti-inflammatory and anti-nociceptive effects in a mouse model of inflammatory pain [116]. Most recently, structural features of known P2 receptor antagonists have been used in the design and development of a thiourea derivative, *N*-((2-bromo-4-isopropylphenyl)carbamothioyl)adamantane-1-carboxamide (Compound 4n), which inhibited recombinant human P2X4 receptor Ca^2+^ responses with >20-fold greater potency than BX430 [117]. Similar to BX430, this compound demonstrated non-competitive, allosteric properties. A summary of the antagonists discussed here and their half-maximal inhibitory concentrations (IC_50_) at the human P2X4 receptor and mammalian P2X4 receptor orthologues, has been provided in Table 2.

## 6. P2X4 Receptors as Molecular Targets in Neuroinflammatory Signalling and Disorders

Neuroinflammation broadly describes an inflammatory response within the central nervous system [118]. Although widely associated with damage to neurons, degradation of neuronal tissue and the onset of neurodegenerative disorders [119,120], positive aspects of the neuroinflammatory response have also been identified, such as injury-induced macrophage remodelling and promotion of axon repair [121]. The neuroinflammatory response is primarily mediated through the activation of microglia and astrocytes, which release pro-inflammatory cytokines, chemokines, and secondary messengers, while endothelia and peripheral immune cells in circulation play an important role in the regulating the crosstalk between immune cell mediators and the central nervous system [118]. Given the P2X4 receptor is present on many of these cell types of the neuroinflammatory axis, there has been a shifting focus towards the role of this receptor in neuroinflammatory signalling and related disorders.

From studies in human and rodent spinal cord microglia, it has been well established that activation of P2X4 receptors by ATP, such as that released from the vesicular nucleotide transporter (VNUT) following peripheral nerve injury in mice [122], stimulates p38 mitogen-activated protein kinase (MAPK) to induce the release of BDNF from microglia [4,24,27]. In a model of neuropathic allodynia, this release of BDNF has been suggested to activate transmembrane tyrosine kinase B on secondary sensory neurons, downregulating the K^+^/Cl^−^ transporter, KCC2, leading to a depolarising shift in the anionic gradient, and the subsequent injury-mediated release of γ-aminobutyric acid (GABA) from interneurons [123]. This dysregulation suppresses transmission through GABAergic inhibitory synapses, leading to an opening of Cl^−^ channels and pain hypersensitivity through disinhibition of the spinal circuits [124]. As such, microglial P2X4 receptors have been suggested to play important roles in the gating of neuropathic chronic pain and tactile allodynia following peripheral nerve injury [7,125,126], traumatic brain injury [127], and in cancer-induced bone pain models [128,129].

Studies have found that P2X4 receptor expression and activation in spinal cord-derived microglia are highly upregulated following peripheral nerve injury [24,130], supporting a role for these receptors in neuropathic pain. This is further supported by findings that P2X4 receptor-deficient mice exhibit reduced responses to chronic neuropathic pain caused by peripheral nerve injury but show no pain response defect to acute noxious stimuli or local tissue damage [49]. In addition to this, P2X4 receptor inhibition with the SNRI duloxetine or the SSRI paroxetine has been demonstrated to reverse neuropathic pain in rats following nerve injury [107,108]. Both of these clinically prescribed anti-depressants have also been used to treat chronic neuropathic pain in humans, although the effectiveness of duloxetine appears to be greater [131,132,133,134,135,136,137] in comparison to paroxetine, which has often demonstrated mixed results regarding effectiveness in reducing chronic pain [138,139,140]. The proposed mechanisms through which these compounds act include signalling through neuronal transient receptor potential channels [141] and the Toll-like receptor 4 signalling pathway [142]. More recently, the inhibition of P2X4 receptors with 5-BDBD has been shown to reduce the inflammasome activation in motoneurons and prevent the loss of these cells in a murine model of sciatic nerve injury [52] indicating that the role of P2X4 receptors in nerve injury extends beyond microglia.

Further to neuropathic pain, P2X4-deficient mice also exhibit reduced responses to chronic inflammatory pain following nerve injury via a similar activation pathway involving p38 MAPK stimulation [49]. This process occurs within tissue-resident macrophages, which have been shown to express P2X4 receptors [3]. The activation of P2X4 receptors in tissue-resident macrophages results in an enzymatic cascade of events, including the activation of cyclooxygenases and the subsequent release of prostaglandin E_2_, a key mediator of inflammation [3]. In addition to this, tissues extracted from P2X4 knockout mice were completely absent of inflammatory prostaglandin E_2_ [3], further supporting the role of the P2X4 receptor in the production of this compound and its signalling of chronic inflammatory pain.

A number of studies have demonstrated that P2X4 receptors may also play a key role in alcohol use disorders [10]. This has been suggested through the upregulation of P2X4 receptors on murine microglial cells, which coincides with increased microglial activation, the release of neuroinflammatory mediators, and the regulation of downstream signalling events [97,143,144]. Furthermore, excessive alcohol use has also been linked with enhanced neuroinflammation [145], promoting microglial activation and peripheral macrophage recruitment into the central nervous system in a murine model of chronic alcohol consumption [146]. Ivermectin, which is known to antagonise the ethanol-mediated inhibition of P2X4 receptors [85], has been assessed in a Phase 1 clinical trial for treating alcohol use disorders [147]. However, despite demonstrating the safety of ivermectin (30 mg taken orally) in combination with an intoxicating dosage of alcohol, this trial was unable to provide evidence for ivermectin in effectively reducing alcohol craving or response to alcohol. Nonetheless, the authors suggest additional studies with larger sampling and alternate dosing regimens are warranted, given the demonstrated safety of ivermectin and its potential for treating alcohol use disorders observed through in vivo studies in mice [148]. Ivermectin has also demonstrated other potential therapeutic benefits outside of alcohol use disorders. Potentiation of P2X4 receptors with ivermectin revealed an anti-bacterial effect and improved survival in a mouse model of sepsis [149], while in a mouse model of multiple sclerosis (autoimmune encephalomyelitis), the potentiation of P2X4 receptors by ivermectin improved oligodendrocyte-mediated remyelination of neurons [150]. In a dopamine depletion mouse model of Parkinson’s disease, the potentiation of P2X4 receptors by ivermectin-enhanced motor behaviour in the presence of levodopa [151], a therapeutic agent used for the treatment of Parkinson’s disease. This demonstrated the potential of ivermectin for the treatment of Parkinson’s disease. Despite these studies, the full effects of ivermectin as a therapeutic agent for neuroinflammatory and neurodegenerative diseases, such as multiple sclerosis and Parkinson’s disease, requires further investigation.

## 7. Conclusions

The P2X4 receptor has emerged as a strong candidate for targeting neuroinflammatory signalling mechanisms, particularly in macrophages, microglia, and endothelial cells. Studies continue to demonstrate links between the P2X4 receptor and important physiological and pathophysiological mechanisms, such as that observed in chronic neuropathic and inflammatory pain models. The continued development of novel drugs targeting the P2X4 receptor over the past decade has contributed to our current understanding of this receptor and its molecular mechanisms. Despite this, the molecular tools currently used to study P2X4 receptors have demonstrated a number of shortfalls, including a lack of selectivity, effectiveness, or low solubility. This has limited the ability to selectively target the physiological effects of the P2X4 receptor, such as in neuroinflammatory models of disease. Although much is left to be understood regarding how best to target these receptors in neuroinflammatory disease, studies continue to deliver promising results in the design and development of effective and selective drugs for elucidating the cellular and molecular properties of the P2X4 receptor. As such, the challenges presented here represent relatively minor roadblocks in the future of targeting the P2X4 receptor as a therapeutic strategy in neuroinflammatory disease.

## Figures and Tables

**Figure 1 ijms-23-05739-f001:**
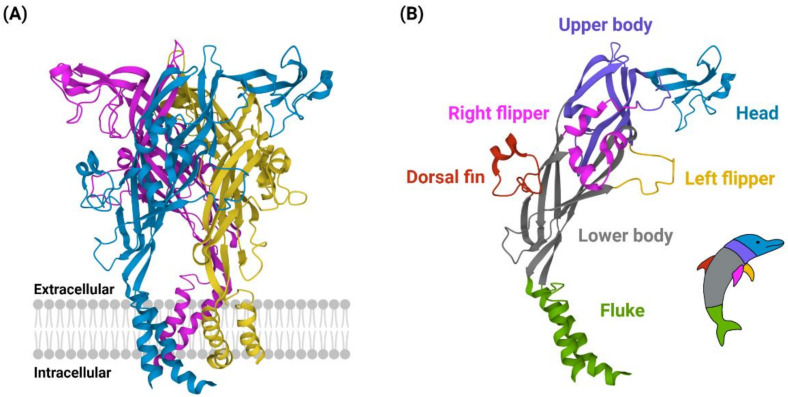
Structural characteristics of the P2X4 receptor. (**A**) Trimeric structure of the gated P2X4 receptor channel in the ATP-bound, open state, depicted as embedded in a cell membrane. Each subunit is labelled a different colour. (**B**) The ‘dolphin-like’ structure of a P2X4 monomer, with each section colour-coded as depicted in the dolphin inset. Structures were reproduced from the RSCB Protein Data Bank file 4DW1 [18] using Mol* Viewer [20]. Created with BioRender.com.

**Figure 2 ijms-23-05739-f002:**
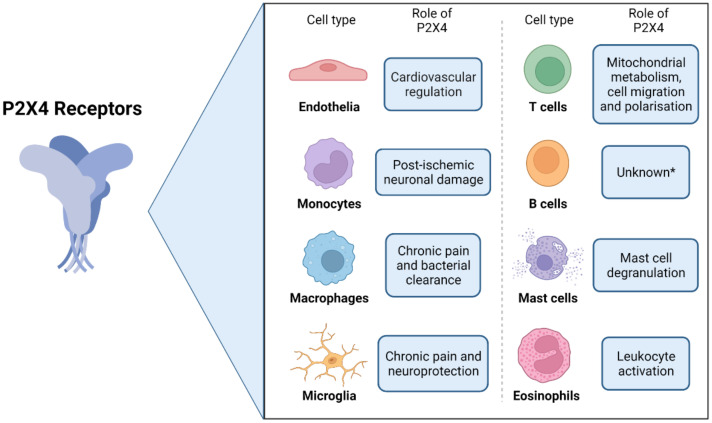
Expression of P2X4 receptors on cells of the neuroinflammatory axis and their proposed cellular or physiological roles. *A role for P2X4 receptors in B cells has not yet been determined. Created with BioRender.com.

**Figure 3 ijms-23-05739-f003:**
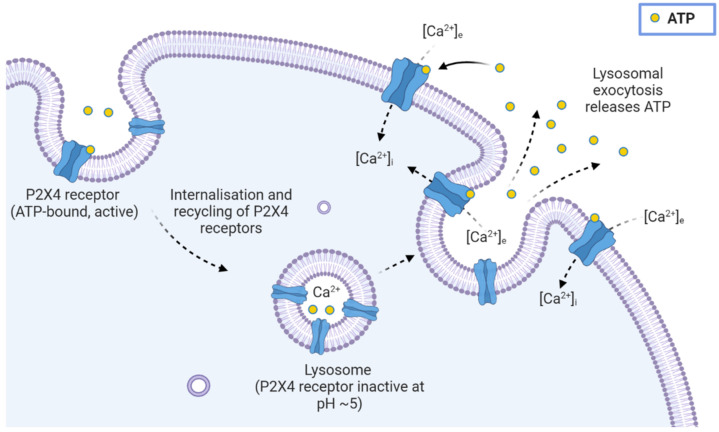
Cell surface and subcellular expression of P2X4 receptors. P2X4 receptors are expressed at both the cell surface, and on lysosomal compartments following internalisation, where their activity is tightly regulated by luminal pH. Endolysosomal recycling, membrane fusion, and exocytosis have been linked with P2X4 receptor activation, autocrine signalling, and Ca^2+^ homeostasis. Abbreviations: ATP, adenosine 5′-triphosphate; [Ca^2+^]_e_, extracellular calcium; [Ca^2+^]_i_, intracellular calcium. Created with BioRender.com.

**Table 2 ijms-23-05739-t002:** Compounds demonstrating antagonistic effects against mammalian P2X4 receptors.

Antagonist	Target ^1^	IC_50_
PPADS	hP2X4	27.5 µM [14]
	rP2X4 ^2^	>500 µM [14]
Suramin	hP2X4	178 µM [14]
	rP2X4 ^3^	>500 µM [14]
Bromophenol blue	hP2X4	78.3 µM [14]
	rP2X4	302 µM [14]
Cibacron blue	hP2X4	39.2 µM [14]
	rP2X4	128 µM [14]
Brilliant Blue G	hP2X4	3.2 µM [95]
	rP2X4	>10 µM [95]
TNP-ATP	hP2X4	1.5 µM [72,104,105], 4.3 µM [69]
	rP2X4	1.3 µM [72], 4.7 µM [105]
	mP2X4	1.3 µM [105], 4.2 µM [72]
	dP2X4	8.1 µM [69]
Paroxetine	hP2X4	1.9 µM [107], 4.8 µM [72], 77.6 µM [69]
	rP2X4	1.6 µM [72], 2.5 µM [107]
	mP2X4	0.7 µM [72]
	dP2X4	13.2 µM [69]
Duloxetine	hP2X4	1.6 µM [108], 17.0 µM [69]
	dP2X4	15.1 µM [69]
NP-1815-PX	hP2X4	0.3 µM [109]
PSB-12054	hP2X4	0.2 µM [105]
	rP2X4	2.1 µM [105]
	mP2X4	1.8 µM [105]
PSB-12062	hP2X4	1.4 µM [105]
	rP2X4	1.8 µM [105]
	mP2X4	0.9 µM [105]
5-BDBD	hP2X4	0.3 µM [72], 1.2 µM [104], 5.2 µM [69]
	rP2X4	3.5 µM [72]
	mP2X4	2.0 µM [72]
	dP2X4	5.8 µM [69]
BX430	hP2X4	0.5 µM [115], 1.9 µM [69]
	dP2X4	7.8 µM [69]
BAY-1797	hP2X4	0.1 µM [116]
	rP2X4	0.1 µM [116]
	mP2X4	0.2 µM [116]
Compound 4n	hP2X4	0.04 µM [117]

^1^ hP2X4, human P2X4; rP2X4, rat P2X4; mP2X4, mouse P2X4; dP2X4, dog P2X4. ^2^ Sensitivity of rat P2X4 receptors to PPADS is restored through Glu249Lys substitution [23]. ^3^ Sensitivity of rat P2X4 receptors to suramin is restored through Gln78Lys substitution [14].

## Data Availability

Not applicable.

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
