# Peer review of "The P2X4 Receptor: Cellular and Molecular Characteristics of a Promising Neuroinflammatory Target"

_ijms, 2022, doi:10.3390/ijms23105739_

Round 1

Reviewer 1 Report

The authors of this review  summarize current knowledge about P2X4 receptor expression, function and  pharmacology in cells involved in neuroinflammatory disorders.  

It is important and well written review, but it can be improved and some points need further clarification. A summary scheme describing type of neuronal cells expressing P2X4 and showing  subcellular localization of P2X in these cells is recommended.

Minor comments

Line 29: instead of  “Activation of P2X4 receptors by ATP results…“ should be   “In cells of immune system, activation of P2X4 receptors by ATP results…“.

Line 101 and  line 110: While the authors in the  line 101 mention ADP  as agonist,  line 110 states that ADP cannot activate mammalian P2X4. This discrepancy should be clarified.

Line 111: Chapter No.4 is followed by chapter No. 6, thus chapter No 5 is missing

Line 225:  “...endogenous P2X4 on  macrophages.” Rodent or human P2X4?  Wherever the species can play a role, it should be mentioned.

Lines 248-251: „This release of BDNF can activate transmembrane tyrosine kinase B on secondary sensory neurons, downregulating the K+/Cl- transporter, KCC2, leading to a depolarising shift in the anionic gradient and the release of γ-aminobutyric acid (GABA) from interneurons” Does it mean that excitatory GABA stimulates its own release from both secondary sensory neurons and  interneurons?

Lines 273-274: „….the role of P2X4 receptors in nerve injury extends beyond microglia“   The authors should consider to include a scheme showing the expression of P2X4 in microglia in relation to other neuronal cells including neurons. Subcellular localization of these receptors (presynaptic, postsynaptic, extrasynaptic?) in neurons could be included. It would increase  the understanding the  role of P2X4  in neuroinflammatory physiology and pathophysiology, which is the aim of this article.

Lines 285-286: „A number of studies have demonstrated that P2X4 receptors also play a key role in 285 alcohol use disorders [10].“ Only one study is mentioned. It is not clear from this review, as well as from the  mentioned paper (Ref 10), what type of P2X4-expressing cells  are most important for alcohol use disorders - neurons or  microglia? Again, a scheme  could provide an answer to this question if it is known.  

Author Response

The authors of this review summarize current knowledge about P2X4 receptor expression, function and pharmacology in cells involved in neuroinflammatory disorders.  

It is important and well written review, but it can be improved and some points need further clarification. A summary scheme describing type of neuronal cells expressing P2X4 and showing  subcellular localization of P2X in these cells is recommended.

Minor comments

  • Line 29: instead of  “Activation of P2X4 receptors by ATP results…“ should be   “In cells of immune system, activation of P2X4 receptors by ATP results…“.

This line has been changed as suggested (line 29; all line numbers refer to revised version).

  • Line 101 and line 110: While the authors in the line 101 mention ADP as agonist, line 110 states that ADP cannot activate mammalian P2X4. This discrepancy should be clarified.

We have added in a statement to explain that traces of ATP in ADP stocks are likely to have contributed to some reports of ADP as an agonist of P2X4 (as we have previously demonstrated [66]) (lines 142-144).

  • Line 111: Chapter No.4 is followed by chapter No. 6, thus chapter No 5 is missing

We have now corrected this (line 150 and chapters/subchapters onwards have been adjusted accordingly).

  • Line 225:  “...endogenous P2X4 on macrophages.” Rodent or human P2X4?  Wherever the species can play a role, it should be mentioned.

We have included the species (“human or canine”) information in this sentence (line 267).

  • Lines 248-251: „This release of BDNF can activate transmembrane tyrosine kinase B on secondary sensory neurons, downregulating the K+/Cl- transporter, KCC2, leading to a depolarising shift in the anionic gradient and the release of γ-aminobutyric acid (GABA) from interneurons” Does it mean that excitatory GABA stimulates its own release from both secondary sensory neurons and interneurons?

Although we cannot say that for sure as would need to be confirmed experimentally, what this paragraph is referring to is the release of GABA from interneurons specifically related to allodynia. These studies together form a hypothesis that has been suggested in a model of allodynia following peripheral nerve injury, as an explanation for pain sensation following a touch stimuli which results in release of GABA (and ATP through VNUT – which has now been added for clarity) from interneurons. The ATP released from interneurons following touch stimuli could stimulate BDNF release from microglia and downstream effects on secondary sensory neurons. Meanwhile GABA released from interneurons following the touch stimuli causes depolarisation of secondary sensory neurons through opening of Cl- channels and subsequent pain sensation.

We have adjusted the language in this paragraph for clarity (lines 307-314).

  • Lines 273-274: „….the role of P2X4 receptors in nerve injury extends beyond microglia“   The authors should consider to include a scheme showing the expression of P2X4 in microglia in relation to other neuronal cells including neurons. Subcellular localization of these receptors (presynaptic, postsynaptic, extrasynaptic?) in neurons could be included. It would increase the understanding the role of P2X4  in neuroinflammatory physiology and pathophysiology, which is the aim of this article.

The expression and distribution of P2X4 receptors in neuronal cells and brain regions is complex and beyond the scope of this review, which primarily focuses on this receptor in non-neural cells, where the function of P2X4 receptors is most heavily documented in neuroinflammatory signalling and disorders. Moreover this has been recently reviewed by Montilla et al. (2020). Thus, we have refrained from implementing this suggestion, but have directed readers to this article (lines 83-88).

  • Lines 285-286: „A number of studies have demonstrated that P2X4 receptors also play a key role in 285 alcohol use disorders [10].“Only one study is mentioned. It is not clear from this review, as well as from the  mentioned paper (Ref 10), what type of P2X4-expressing cells  are most important for alcohol use disorders – neurons or  microglia? Again, a scheme could provide an answer to this question if it is known.  

P2X4 receptor-expressing murine microglia have been implicated in disorders related to excessive alcohol consumption. We have now added further information to this section in line with the reviewer’s comments (lines 349-355).

Reviewer 2 Report

Comments for manuscript: “ The P2X4 Receptor: Cellular and Molecular Characteristics of a Promising Neuroinflammatory Target”, ijms-1684155.

This is an interesting review aimed at describing the potential of P2X4 receptor as mediator of neuroinflammation, the content is well organized and of general interest. I have only some commentaries

Commentaries

1.- In the section “Introduction” structural information on monomeric P2X4 receptors is exposed, this reviewer considers that a figure showing a schematic representation of the subunit and/or the channel can enrich the manuscript.

2.- Information on the pharmacology of P2X4 could also be presented in a table, this summarized format is useful for the readers.

3.-In section 7 “P2X4 receptors as molecular targets in neuroinflammatory signalling and disorders”, basic concepts on neuroinflammation should be included: What is neuroinflammation, mediators of inflammation and cells related with immune response in the brain.

Author Response

This is an interesting review aimed at describing the potential of P2X4 receptor as mediator of neuroinflammation, the content is well organized and of general interest. I have only some commentaries

Commentaries

  • In the section “Introduction” structural information on monomeric P2X4 receptors is exposed, this reviewer considers that a figure showing a schematic representation of the subunit and/or the channel can enrich the manuscript.

We have included a figure demonstrating the trimeric channel as well as monomeric subunit (new Figure 1) as described in-text (lines 43 and 47).

  • Information on the pharmacology of P2X4 could also be presented in a table, this summarized format is useful for the readers.

We have consolidated information on the pharmacology of the P2X4 receptor, as requested, in the form of two tables. Firstly, Table 1 summarising the reported potencies ATP at the P2X4 receptor; and secondly, Table 2 summarising the reported potencies of antagonists of the P2X4 receptor.

  • In section 7 “P2X4 receptors as molecular targets in neuroinflammatory signalling and disorders”, basic concepts on neuroinflammation should be included: What is neuroinflammation, mediators of inflammation and cells related with immune response in the brain.

We have included a paragraph to include this information on neuroinflammation to give further context to this section (lines 295-306), which includes the addition of four new references [118-121].